# Linearolactone Induces Necrotic-like Death in *Giardia intestinalis* Trophozoites: Prediction of a Likely Target

**DOI:** 10.3390/ph15070809

**Published:** 2022-06-29

**Authors:** Raúl Argüello-García, Fernando Calzada, Bibiana Chávez-Munguía, Audifás-Salvador Matus-Meza, Elihú Bautista, Elizabeth Barbosa, Claudia Velazquez, Marta Elena Hernández-Caballero, Rosa Maria Ordoñez-Razo, José Antonio Velázquez-Domínguez

**Affiliations:** 1Departamento de Genética y Biología Molecular, Centro de Investigación y Estudios Avanzados, IPN, Mexico City 07360, Mexico; raularguellogarcia@yahoo.com; 2Unidad de Investigación Médica en Farmacología, 2° Piso CORCE, UMAE Hospital de Especialidades, Centro Médico Nacional Siglo XXI, Av. Cuauhtémoc 330, Colonia Doctores, Mexico City 06720, Mexico; 3Departamento de Infectómica y Patogénesis Molecular, CINVESTAV-IPN, Av. Instituto Politécnico Nacional No. 2508, Colonia San Pedro Zacatenco, Mexico City 07360, Mexico; bchavez@cinvestav.mx; 4Department of Pharmaceutical Sciences, College of Pharmacy, University of Nebraska Medical Center, 985830 Nebraska Medical Center, Omaha, NE 68198, USA; audi_matus@hotmail.com; 5CONACyT-División de Biología Molecular, Instituto Potosino de Investigación Científica y Tecnológica A. C., San Luis Potosí 78216, Mexico; francisco.bautista@ipicyt.edu.mx; 6Sección de Estudios de Posgrado e Investigación, Escuela Superior de Medicina, Instituto Politécnico Nacional, Salvador Díaz Mirón Esq. Plan de San Luis S/N, Miguel Hidalgo, Casco de Santo Tomas, Mexico City 11340, Mexico; rebc78@yahoo.com.mx; 7Área Académica de Farmacia, Instituto de Ciencias de la Salud, Universidad Autónoma del Estado de Hidalgo, Km 4.5, Carretera Pachuca-Tulancingo, Unidad Universitaria, Pachuca 42076, Mexico; cvg09@yahoo.com; 8Facultad de Medicina, Biomedicina, Benemérita Universidad Autónoma de Puebla, Puebla 72410, Mexico; ehdezc@yahoo.com; 9Unidad de Investigación Médica en Genética Humana, UMAE Hospital de Pediatría 2° Piso, Centro Méddico Nacional Siglo XXI, Instituto Mexicano del Seguro Social, Av. Cuauhtémoc 330, Colonia Doctores, Mexico City 06725, Mexico; 10Escuela Nacional de Medicina y Homeopatía IPN, Av. Guillermo Massieu Helguera, No. 239, La Escalera, Mexico City 07320, Mexico

**Keywords:** *neo*-clerodane diterpenes, *Giardia intestinalis*, ROS production, ultrastructural alterations, docking analysis, aldose reductase

## Abstract

Linearolactone (LL) is a *neo*-clerodane type diterpene that has been shown to exert giardicidal effects; however, its mechanism of action is unknown. This work analyzes the cytotoxic effect of LL on *Giardia intestinalis* trophozoites and identifies proteins that could be targeted by this active natural product. Increasing concentrations of LL and albendazole (ABZ) were used as test and reference drugs, respectively. Cell cycle progression, determination of reactive oxygen species (ROS) and apoptosis/necrosis events were evaluated by flow cytometry (FCM). Ultrastructural alterations were analyzed by transmission electron microscopy (TEM). Ligand–protein docking analyses were carried out using the LL structure raised from a drug library and the crystal structure of an aldose reductase homologue (GdAldRed) from *G. intestinalis*. LL induced partial arrest at the S phase of trophozoite cell cycle without evidence of ROS production. LL induced pronecrotic death in addition to inducing ultrastructural alterations as changes in vacuole abundances, appearance of perinuclear and periplasmic spaces, and deposition of glycogen granules. On the other hand, the in silico study predicted that GdAldRed is a likely target of LL because it showed a favored change in Gibbs free energy for this complex.

## 1. Introduction

The protozoan *Giardia intestinalis* (syn. *G. lamblia* and *G. duodenalis*) is the causative agent of giardiasis, a cosmopolitan parasitosis of great epidemiological and clinical importance, which occurs with high incidence in children [1]. This infection is transmitted by a fecal–oral route through contact with infected people or companions, livestock and wild animals, or by consumption of water or food contaminated with cysts [2,3]. This parasitosis can be asymptomatic, although as part of its signs it can produce chronic diarrhea and fainting [3,4]. Furthermore, giardiasis has been associated with impaired physical and cognitive development in children [5,6,7]. The administration of antiparasitic drugs such as metronidazole has been shown to cause undesirable side effects as well as resistance mechanisms in some protozoa, which contributes to the need to search for new, more effective and safer agents with a selective toxicity for the parasitic organism and minimal effects on the host [8,9,10].

Nowadays, the specialized metabolites derived from plants such as flavonoids and terpenoids are considered to be promising scaffolds for the development of antiparasitic agents [8,9,10]. In Mexico, plants belonging the *Lamiaceae* family are a rich source of these kind of compounds, particularly the species from *Salvia* genus. Linearolactone (LL) is a furane diterpenoid of the *neo-*clerodane framework that is obtained from the aerial parts of *Salvia polystachya* [11,12]. This compound has been reported as one of the active constituents of the amoebicidal effect, and as a stimulation in the expression of the extracellular matrix components in wounding processes attributed to the plant. In addition, a previous in vitro study reported the antigiardial activity of LL. However, so far, its giardicidal molecular mechanisms of action remain unknown [11,12,13,14,15]. The present work aimed to characterize the giardicidal mechanism of LL (Figure 1) as a likely molecular target underlying the effects of this compound.

## 2. Results

To analyze the effect of LL on *G. intestinalis,* the distribution of cells at cell cycle stages (G0/G1, S, G2-M) was determined using trophozoites, showing distinct polyploidy in the culture due to DNA replication (4N and 8N). We believe that this process could be altered by LL. In these experiments, protozoan DNA was colored with PI, and RNA was digested with RNase A in samples exposed to multiple concentrations of LL (CI: 50–200). Figure 2 shows the histograms of these assays. In the culture with exposure to DMSO, the G0/G1 protozoa subpopulation was significant (78.8%), followed by cells in the S phase (13%) and a small proportion of trophozoites at the G2/M phases (5.6%) (Figure 2a). The effect of the four concentrations of LL assayed on *G. intestinalis* trophozoites displayed a dose-dependent increment in cells in the S phase (13–24%), with a corresponding decrease in the G0/G1 subpopulation (68–78%), and the G2/M subpopulation displayed minimal changes (2–7%). In addition, the trophozoites in the S phase displayed a partial arrest. Thereby, the above data indicate that LL allows the G0/G1→S transit in the cycle cell of *G. intestinalis*; the lack of significant changes in the G2/M protozoa populations may be caused by a cytostatic effect anteceded by the narrow previous nuclear DNA replication that in turn does not allow more cytokinesis.

To evaluate whether LL has the capability to produce reactive oxygen species (ROS) in *G. intestinalis* trophozoites, due to the presence of lactone groups and a furan ring in the structure of LL, the ROS-sensitive fluorescent marker DCH2FDA was used to determine the ROS production. For this purpose, the positive controls tert-butyl hydroperoxide (TBHP) and albendazole (ABZ) were used. As can be seen in Figure 3c,d (TBHP and ABZ treatments, respectively), the treatments with positive controls caused oxidative stress according to other previously reported studies [16,17,18]. Thus, the ROS generation using TBHP caused a significant change in cellular fluorescence in the order of up to 76% (Figure 3c). In contrast, the treatments with LL at CI_50_ and CI_100_ (Figure 3e,f, respectively) did not produce a displacement of cellular fluorescence as occurred with TBHP and ABZ and suggests that LL does not have the capability to produce ROS. Therefore, it is worth noting that the giardicidal mechanism of action of LL is different to the mechanism of ABZ, which can also be used for the positive control of cell damage.

The effects of LL on the ultrastructure of *G. intestinalis* trophozoites were further analyzed by TEM. In panel A of Figure 4 (Figure 4a,c; magnified in Figure 4b–d) (untreated cells and in the presence of the vehicle DMSO, respectively), it is shown that the ultrastructure of the trophozoites are intact. This evidenced a structured endomembranous system in the cytoplasm, with the pleomorphic peripheral vacuoles arrangement typical of this parasite (PV). This comprised rounded nuclei (N) with the electronic density of heterochromatin and the proper elements of a cytoskeleton, including flagellar axonemes (A) with a 9 + 2 matrix of microtubules and microtubules of ventral adhesive disc (AD); in a cross-sectional view, these show a head and tail arrangement (Figure 4b, inset). When the trophozoites were exposed to ABZ, a compound that acts directly on the microtubules [19], ultrastructural alterations were marked, displaying partial destruction of the endomembranous tissue (Figure 4e), with a clear loss of AD microtubules (Figure 4c, * panels Figure 4e,f; Figure 4g,h). In a cross-sectional view, these show a head and tail arrangement (Figure 4f), as well as the loss of organization of AD microtubules (Figure 4h, red arrowhead). In trophozoites treated with LL, ultrastructural alterations were also observed, such as the appearance of large, irregularly shaped perinuclear and periplasmic spaces (≈1 µm) devoid of electron-dense content (Figure 4h, arrowhead). In most cases, these were adjacent to areas bordered by membranes with deposits of electron-dense granules (Figure 4g, arrowhead). Additionally, treatment with LL induced a decrease in peripheral vesicles (PV), thereby favoring the presence of glycogen granules arranged (EDG) as part of the cytoplasm. It is important to note that neither the ABZ nor the LL treatments provoked alterations in the nuclei that were otherwise preserved. Likewise, flagellar axonemes (A) and the adhesive disk appear unchanged by LL treatment. In addition, the cytotoxic effect of LL on *G. intestinalis* trophozoites was identified as a possible direct cell death (necrosis-like) preceded by a programmed cell death (apoptosis-like). In this sense, annexin V-FITC was used for these studies as a marker to monitor phosphatidylserine exposure in the outer protozoa membrane, showing early or late apoptosis (zones R4 and R2 in the histograms of Figure 5) and PI was used as a dye that directly stains DNA and cytoplasmic components due to direct damage to the cell membrane (region R1). Results are shown in histograms and average counts included in each region. As compared with untreated cells (Figure 5a), DMSO caused a very limited increase in region R2 (3–11%; Figure 5b), (R3 region, Figure 5c). Interestingly, the exposure of trophozoites with LL at IC_25_ (Figure 5c), IC_50_ (Figure 5d) and IC_100_ (Figure 5e) did not induce significant increases in parasite numbers in early apoptosis (1.7–2.2%: region R4). Likewise, apoptotic cell numbers were almost maintained (11.8–9.6%; region R2). However, a significant concentration-dependent increase in cells at necrosis was noteworthy (≈2, 15 and 45%, respectively; region R1). From these data, a necrotic-like death in trophozoites upon exposure to LL was shown.

In a review of studies of enzyme inhibition by terpene compounds, it was found that several members of the terpene family, such as ginsenosides (triterpene saponins), perillosides A and C (monoterpene glycosides) and triterpenes/meroterpenes from *Ganodema lucidium* (lanostane and farnesyl hydroquinone-triterpene conjugates), have the ability to inhibit AldRed from humans and other species [18,19,20,21]. This prompted us to predict whether the diterpene LL was able to interact and inhibit GdAldRed using molecular docking approaches. In these analyses, the crystal structure of GdAldRed (PDB ID: 3KRB) was docked with sorbinil, a well-known AldRed inhibitor with clinical applications in diabetes complications [22] and LL; in addition, a “blind” docking was carried out with LL to show its most favored docking against the GdAldRed dimer. Results of these predictions are shown in Figure 6. As can be seen in Figure 6A, LL docks with a most favored position (ΔG = −8.05 kCal/mol) just at the dimer interface, which could be important to promote dimer dissociation with further inactivation of the enzyme. When the docking with LL within the catalytic site of GdAldRed was analyzed, it was observed that LL preserves one of the relevant interactions for the inhibitory activity, i.e., between the ring of the range of lactones with the hydrogen of the indoline of Trp105 (Figure 6b). In addition, hydrogen bonding of the gamma lactone amine oxygen of the NADP nicotinamide was observed, which could suggest a greater effect on the inhibitory activity (Figure 6d). Sorbinil (AldRed inhibitor) was used as internal control, showing a strong affinity for the active site (ΔG = −7.7 kcal/mol). Remarkably, the ΔG values shed by these studies suggest that LL has a better ability to bind GdAldRed than the reference inhibitor sorbinil. From these data, it is predicted that GdAldRed is a viable target of LL and that LL presented a better binding affinity in the active site.

## 3. Discussion

LL is a diterpene that was first isolated from *Salvia lineata* [23] and later from *S. polystachya* [11]. Previous studies have strongly evidenced its antiprotozoal properties, including giardicidal activities [24]. So far, there are no reports describing its giardicidal mechanism of action; hence, the present work provides evidence that LL exerts cytotoxic damage at the cellular and molecular level in *G. intestinalis*. The results obtained in this study indicated a partial arrest at the S phase of the cell cycle in trophozoites treated with increasing concentrations of LL. These results also allowed us to infer the possibility that LL is able to block DNA synthesis in the parasite to promote a likely spontaneous type of cell death. According with this latter notion, reactive oxygen species (ROS) were not generated in trophozoites by exposure to LL, unlike that observed in cells committed to a programmed cell death mechanism, such as apoptosis [25,26]. As shown in Figure 3c,d, TBHP (a known inducer of ROS) and ABZ (a first-line antigiardial drug) effectively induced ROS formation at concentrations of 100 µM and 0.48 µM, respectively. These data clearly suggest that not only do LL and ABZ have mechanisms of action that are strikingly different on *Giardia* trophozoites, but that direct damage on trophozoite structures or molecules might be driving the cell death mechanism.

At the trophozoite ultrastructure level, studies carried out with other terpenes (incomptine A) have reported alterations at the level of the cytoskeleton, decreases in the presence of peripheral vacuoles and increases in glycogen granules in parasites such as *Entamoeba histolytica* [10]. Other studies, with diterpenes such as (5R, 8R, 9S, 10R)-12-oxo-ent-3, 13(16)-clerodien-15-oic acid and 16-hydroxy-clerod-3,13(14)-diene-15,16-olide, have been able to effect ultrastructural alterations in protozoan parasites such as *Leishmania mexicana* [27]. Considering that the diterpene LL could induce ultrastructural alterations in *G. intestinalis* trophozoites, TEM analyses showed that LL induced alterations such as the important loss of peripheral vesicles (PV) and alterations such as the appearance of irregularly shaped perinuclear (PN) and periplasmic (PPS) spaces, devoid of electron-dense content and areas outlined by membranes with deposition of electron-dense granules (EDG).

As far as the cell death mechanism induced by LL in trophozoites is concerned, Annexin V-PI staining with the use of this diterpene at IC_50_ and IC_100_ suggest that necrosis, a non-programmed cell death mechanism, was promoted (Figure 5B). This finding is in good agreement with previous observations that LL causes DNA synthesis blocking without ROS production (Figure 2 and Figure 3) and supports the presence of a distinct cytotoxic mechanism exerted by LL as compared with other giardicidal compounds, such as the drug ABZ and the flavonoid, kaempferol, that kills trophozoites by apoptosis-like processes [16,17]. The molecular characterization of this necrotic-like process deserves further study, especially for the identification of structures or molecules targeted by LL in this protozoan parasite.

In light of the results obtained herein, identifying a likely target of LL in Giardia with precision is difficult. However, with the presence of electron-dense granules (likely containing glycogen) arranged in the cytoplasm of *G. intestinalis* trophozoites, it is conceivable that the glucose-dependent energy metabolism of this parasite could have been altered. In this scenario, two main pathways managing glucose could be involved, namely glycolysis and the polyol pathway, of which AldRed is a rate-limiting enzyme catalyzing the glucose→sorbitol conversion, and is present in *G. intestinalis* [28,29]. In addition, a literature survey allowed the support of a possible interaction of LL and GdAldRed since several kinds of terpenes are effective AldRed inhibitors [19,20]. The use of bioinformatics tools such as molecular docking in “blind” and “active site-directed” modes allowed us to predict that GdAldRed is a viable target of GdAldRed through two proposed mechanisms: by dimer dissociation (Figure 6A) and active-site inhibition (Figure 6B). Furthermore, our in silico data suggest that LL could be a better inhibitor of GdAldRed than sorbinil is and even an association between these interactions and the appearance of electron-dense granules cannot be ruled out. With regard to the amino acids involved in the predicted interactions of LL with the active site of GdAldRed, two bioactive anti-diabetic compounds (astaxanthin and zeaxanthin) bind to the same site (Trp20 and TYr48) as LL on the catalytic site with AldRed [30]. Other studies carried out in mammalian podocytes under conditions of osmolarity and high glucose concentrations showed a considerable increase in the expression of AldRed; however, it was not accompanied by the activity of the enzyme [31]. According to previous reports, the key amino acid residues for inhibitory activity are Tyr48, His110, Lys77, Trp111, Trp20 and Lys21 in the human AldRed [30] (PDB ID: 1AH0), in which *G. lamblia* corresponded to Tyr40, His104, Lys71, Trp105, Trp12 and Gln13 (Figure 6d). Nevertheless, the presence of a necrotic-like cell death opens the possibility that other molecules are involved as direct targets of LL in this parasite.

## 4. Materials and Methods

### 4.1. Isolation and Purification of LL

From the aerial parts of *Salvia polystachya*, the active compound LL was isolated, purified and characterized by chromatographic and nuclear magnetic resonance (NMR) methods as previously reported, obtaining a purity of ˃95%.

### 4.2. Cell Culture and Exposure to LL

The trophozoites of *G. intestinalis* (strain WB) were maintained in axenic culture in modified Diamond’s TYI-S-33 medium [13] containing bovine bile (Sigma-Aldrich, St. Louis, MO, USA; 0.5 mg/mL) at 37 °C in 4.5 mL screw-cap flat-bottom vials or 15 mL tubes (Falcon^TM^). The parasites were harvested in logarithmic phase of growth (after 72–96 h) by cooling vials or tubes for 45 min, washed three times with phosphate-buffered saline (PBS), counted in a hemocytometer and the protozoan samples were adjusted. To test the properties of LL on trophozoites, four inhibitory concentrations (ICs) were used, 50, 100, 150, and 200, based on the previously reported IC_50_ value of 28.2 μM [11]. A total of 6000 trophozoites/mL were incubated for 48 h/37 °C in the presence of LL previously diluted in vehicle (dimethyl sulfoxide (DMSO, Sigma-Aldrich, St. Louis, MO, USA); 0.022% final concentration). As a positive control for damaged cells, albendazole (ABZ, Sigma-Aldrich, St. Louis, MO, USA) was used at 0.32 μM.

### 4.3. TEM 

Once the trophozoites were exposed with LL at IC_100_ (56.4 µM), they were treated three times with PBS. Trophozoites pellets were fixed with glutaraldehyde and then stained with 1% osmium tetroxide (Sigma-Aldrich, St. Louis, MO, USA) in cacodylate buffer. Parasites were then dehydrated with increasing concentrations of ethanol (from 10 to 90%) for 10 min, followed by treatment with propylene oxide: alcohol mixtures (2:1, 1:1, and 1:2) for 15 min. Pre-embedding was performed using propylene oxide: epoxy resin (2:1, 1:1, and 1:2) for 15 min. Finally, polymerization was carried out by incubation at 60 °C for 24 h. Thin sections of 70 nm were cut and mounted on copper grids. Sections were stained with uranyl acetate (Sigma-Aldrich, St. Louis, MO, USA) for 15 min and lead citrate for 20 min (both from SP1 Supplies; SP1-Chem, West Chester, PA, USA). Thin sections (approximately 30 nm thick) were observed in a JEOL 100-SX transmission electron microscope.

### 4.4. Cell Cycle Analysis

To determine the proportions of trophozoites in different stages of the cell cycle (G0/G1, S and G2/M) after exposure to LL at IC (50, 100, 150 and 200), the nuclei were stained with PI (Thermo Fisher Science, Waltham, MA, USA). Protozoa were treated with PBS, then fixed in ethanol (70%) and incubated at 4 °C in RNase A (0.1 mg/mL). Finally, PI was added and then the samples were quantified in an FACS Calibur^TM^ FCM with a filter at 585 nm and parasite cycle phases were identified in the histogram regions as previously reported [15]. At least 25,000 cells per sample were recorded in these assays (three independent experiments).

### 4.5. Detection of Reactive Oxygen Species (ROS) Using DCH2FDA

Trophozoites were untreated or exposed to vehicle DMSO and LL at IC (50 and 100), washed with PBS and incubated for 30 min at 37 °C with the fluorescent marker 2′, 7′-dichloro-dihydroxy-fluorescein diacetate (DCH2FDA, (Thermo Fisher Science, Waltham, MA, USA) at 25 μM (Image kit -IT^TM^ Live Green ROS Detection; Life Technologies, Carlsbad, CA, USA). Finally, parasites (25,000 per sample; 3 independent experiments) were quantified on a FACSCalibur^TM^ flow cytometer using a 530 nm filter.

### 4.6. Identification of the Type of Cell Death Induced by LL

Trophozoites were exposed to IC (25, 50 and 100) of LL under the aforementioned conditions (Section 4.2). Programmed (apoptosis-like) or spontaneous (necrosis-like) cell death was determined using an anti-annexin V antibodies–fluorescein isothiocyanate (FITC) conjugate and propidium iodide as markers by a commercial kit (Bio Vision, Waltham, MA, USA) following the manufacturer’s instructions. Trophozoites were washed with PBS and incubated for 5 min at room temperature in 400 μL of PBS containing 5 μL each of annexin V and propidium iodide solutions. Fluorescence in cells was quantified in at least 25,000 cells (*n* = 3) using a FACS Calibur^TM^ flow cytometer (Becton Dickinson^TM^) equipped with filters at 530 nm (FITC, FL1) and 585 nm (PI, FL2).

### 4.7. Molecular Docking Study

The preparation of the receptors and ligands was carried out using the AutoDockTools program. The ligand, water molecules and hetero groups were removed from the receptor, with which their crystalline structures were resolved. Subsequently, a grid box of 40 × 40 × 40 points with a grid spacing of 0.375 Å was calculated for the proper atom types and centered on the coordinates 25.04, 39.803 and 14.231. The structures were built in Spartan’10 and subjected to a conformational search using molecular mechanics with MMFF as the force field. The lowest energy conformal was taken as the starting structure for the geometric optimization by means of DFT with functional hybrid B3LYP and 6-31G** basis 1. Afterward, using AutoDock 4.2.6., LL and Sorbinil (SOR) were docked inside the active site of aldose reductase of *G. intestinalis* (GdAldRed) complexed with NADP (PDB code: 3KRB). Visualization of molecular coupling results was carried out in UCSF chimera 1.13.1 to identify the type of interactions of enzyme´s amino acids with both ligands [17]. To assess the interaction of LL (ligand) with the dimer of GdAldRed (target), which was the catalytically competent entity, a “blind docking” analysis was carried out using the server Swissdock (https://swissdock.ch/docking) (accessed on 15 June 2022). This platform retrieves 256 viable docking ligand–target complexes, from which the most favored (lowest delta G energy) was employed for visualization and edition using Chimera V10.1.1.

## 5. Conclusions

The present study used cytotoxic effects of LL on *G. intestinalis* including cell cycle progression, determination of reactive oxygen species (ROS), apoptosis/necrosis events and ultrastructural alterations, followed by the molecular docking approach to determine its potential as an antiparasitic agent. Our findings suggest that LL is a *neo*-clerodane type diterpene with antigiardial potential that induces alterations at the ultrastructural level, such as an important decrease in peripheral vesicles and an increase in electron-dense granules in *G. intestinalis* trophozoites, by a pronecrotic mechanism involving arrest at S phase and absence of ROS. In addition, the molecular docking study suggested that antigiardial effects may be explained in part by the affinity of LL to glycolytic enzyme GdAldRed. Future systematic works will investigate how LL regulates the expression of glycolytic enzyme GdAldRed and its association with cell cycle, apoptosis/necrosis events and cytoskeleton alterations.

## Figures and Tables

**Figure 1 pharmaceuticals-15-00809-f001:**
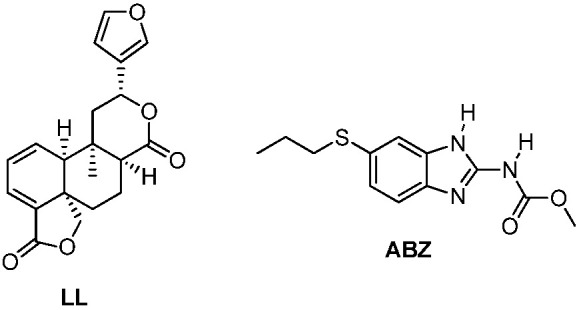
Structure of linearolactone (**LL**) and albendazole (**ABZ**).

**Figure 2 pharmaceuticals-15-00809-f002:**
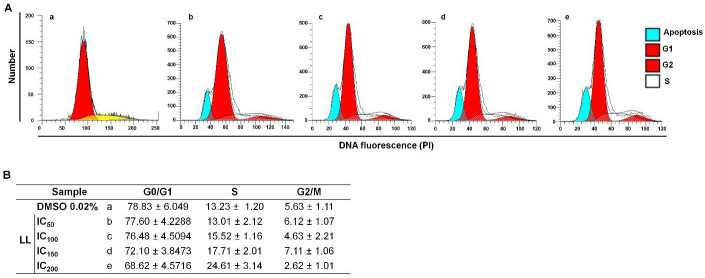
Linearolactone (LL) causes a partial arrest at the S phase in *Giardia intestinalis* trophozoites. (**A**) Trophozoites of strain WB were exposed to vehicle [0.02% dimethyl sulfoxide (DMSO)] (**a**) and LL at IC_50_ (**b**), IC_100_ (**c**), IC_150_ (**d**), IC_200_ (**e**), as shown in the corresponding histograms. The trophozoites were treated with RNase A, the nuclei were colored using PI and processed using flow cytometry (FCM). Peaks of the different cell cycle phases are shown at the top right of the graphs. (**B**) Percentages of cells in each cell cycle stage in the samples described above. The results correspond to the mean ± standard deviation of three independent experiments.

**Figure 3 pharmaceuticals-15-00809-f003:**
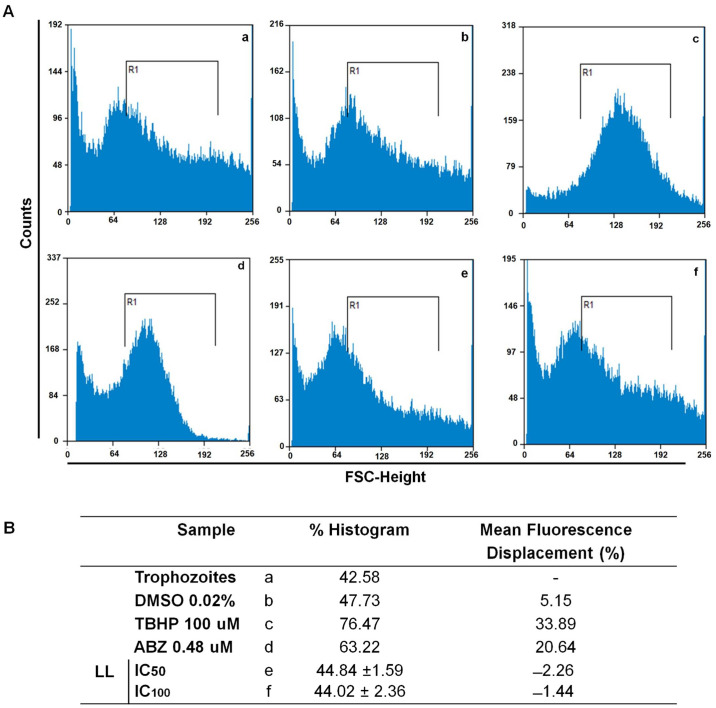
Linearolactone (LL) does not cause oxidative stress in *G. intestinalis* trophozoites. (**A**) Strain WB trophozoites were cultured in TY1-S-33 medium (**a**), exposed to vehicle [0.02% dimethyl sulfoxide (DMSO) (**b**), or exposed to TBHP (100 µM) (**c**) and albendazole 0.48 µM (ABZ) (**d**), LL at IC_50_ (**e**) and IC_100_ (**f**), incubated with dichloro-dihydro fluorescein diacetate to detect ROS. Fluorescence was analyzed by FCM [X-axis: fluorescence scatter (FSC); Y-axis: protozoa count]. R1 showed the ROS fluorescence for untreated and DMSO-treated protozoa. (**B**) The percentages of trophozoites with different fluorescence are shown. The results are expressed as the mean ± standard deviation (*n* = 3).

**Figure 4 pharmaceuticals-15-00809-f004:**
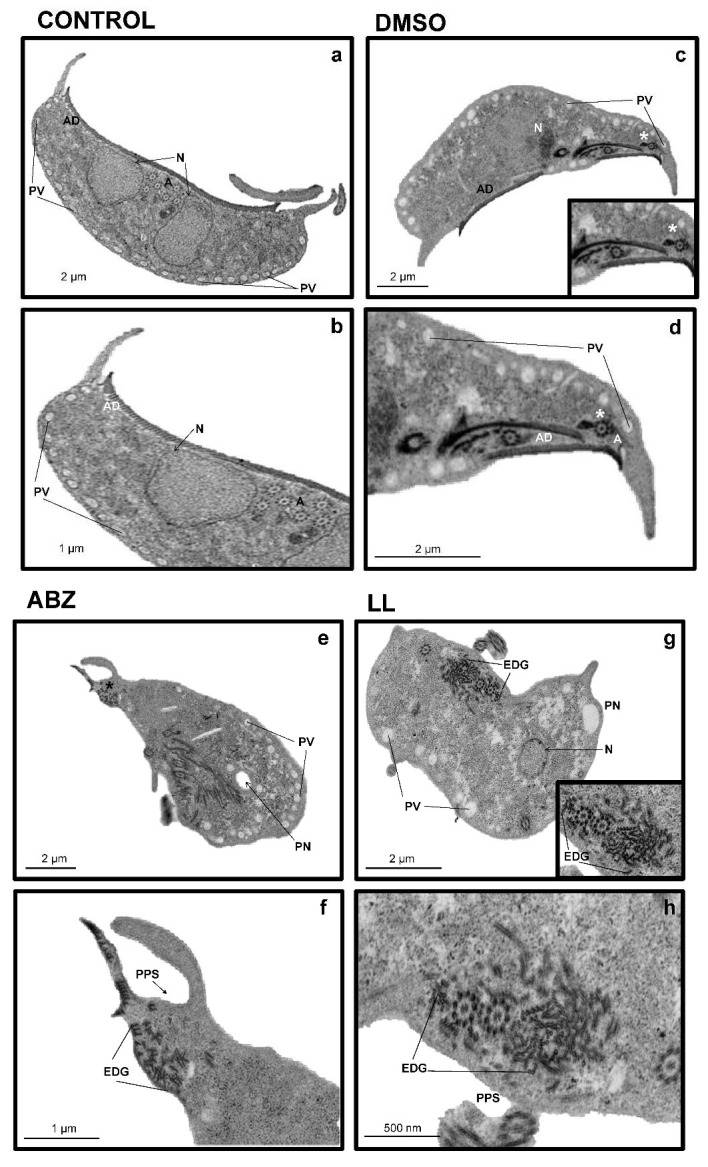
Ultrastructural and necrotic effect of Linearolactone (LL) on *G. intestinalis* trophozoites. Ultrastructural effect of LL in *G. intestinalis* trophozoites. Strain WB trophozoites (6000/mL) were cultured in TY1-S-33 medium (**a**,**b**) or exposed 40 h/37 °C to vehicle cal [0.02% dimethyl sulfoxide (DMSO)] (**c**,**d**) as negative control, 0.32 µM albendazole (ABZ) as a positive control for damage to the cytoskeleton (**e**,**f**) and LL the IC_100_ (**g**,**h**) and processed for transmission electron microscopy (TEM). Each panel shows the scale (bar). A: flagellar axonemes; AD: adhesive ventral disc; N: core; PV: peripheral vesicles * AD Fragments; PN: perinuclear space; PPS: periplasmic spaces; EDG: deposited electron-dense granules.

**Figure 5 pharmaceuticals-15-00809-f005:**
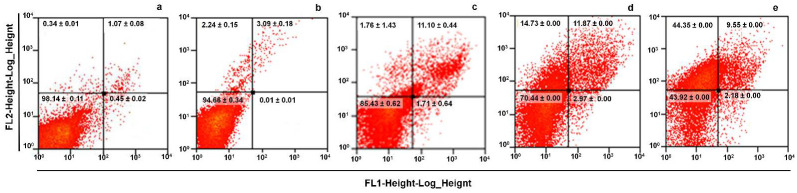
Necrotic effect of linearolactone LL on *G. intestinalis* trophozoites. Strain WB trophozoites were untreated (**a**), treated with DMSO (**b**) and exposed to LL at IC_25_ (**c**), IC_50_ (**d**) and IC_100_ (**e**) (then colored with annexin V-FITC/PI and analyzed by FCM. Histograms are shown (R1 = necrosis; R2 = like late apoptosis; R3 = unstained protozoa; R4 = early apoptosis).

**Figure 6 pharmaceuticals-15-00809-f006:**
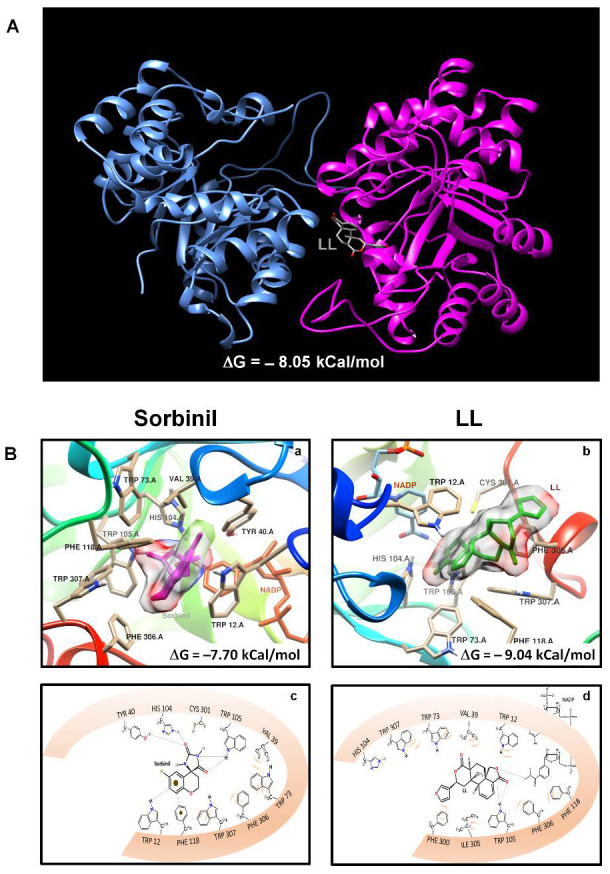
Linearolactone (LL) is a diterpene with a high affinity for aldose reductase from *G. intestinalis* (GdAldRed). (**A**) Molecular docking performed in “blind” mode with LL dimer displayed in stick conformation (blue and red). The free Gibbs energy value (ΔG) is indicated. (**B**) Molecular coupling of the active site of GdAldRed with sorbinil (GdAldRed inhibitor) (**a**) and LL (**b**). The affinity and bonds established by the molecular coupling with GdAldRed with sorbinil (**c**) and LL (**d**). The ΔG data are expressed in kCal/mol.

## Data Availability

Data is contained within the article.

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
