# Peer review of "Linearolactone Induces Necrotic-like Death in Giardia intestinalis Trophozoites: Prediction of a Likely Target"

_pharmaceuticals, 2022, doi:10.3390/ph15070809_

Round 1

Reviewer 1 Report

The manuscript by Argüello-García et al. is very complex, well written and conducted, with important contribution to the understanding of giardicide mecanisms of Linearolactone. I support its possible publication after appropriate minor modifications as outlined below.

Line 52: “Giardiasis” – lowercase

Line 53: please insert a reference after the first sentence

Line 66: please insert a reference after the sentence

Line 272: “NMR” – please explain the acronym

Line 276: “(Sigma Chem...” – for all of the reagents and throughout the manuscript please provide the full name of the company, city and country

Line 349: within the conclusion section the authors must to highlight the study limitation and future perspectives, indicating new research directions in this area

The reference list not completely fulfill the journal requirement. Please carefully revise it!

Author Response

Reviewer 1

Comments and Suggestions for Authors

The manuscript by Argüello-García et al. is very complex, well written and conducted, with important contribution to the understanding of giardicide mechanisms of Linearolactone. I support its possible publication after appropriate minor modifications as outlined below.

Answer

Thanks, reviewer, for your comments and suggestions in agree with your suggestion we made the all-minor modifications; these are in yellow color.

Query 1: Line 52: “Giardiasis” – lowercase

Answer 1: Giardiasis was changed as “giardiasis” 

Query 2: Line 53: please insert a reference after the first sentence

Answer: The reference [1,2] was changed to the first sentence as “ …. with high incidence in children [1].”

Query 3: Line 66: please insert a reference after the sentence

Answer: A reference was insert after sentence as ….. the aerial parts of Salvia polystachya [11-12].

Query4: Line 272: “NMR” – please explain the acronym

Answer: The acronym was explained as nuclear magnetic resonance (NMR)

Query 5: Line 276: “(Sigma Chem...” – for all of the reagents and throughout the manuscript please provide the full name of the company, city and country

Answer: company, city and country were included for all of reagents

Query 6:  Line 349: within the conclusion section the authors must to highlight the study limitation and future perspectives, indicating new research directions in this area

Answer: Conclusion was restructured and additional text was included:

The present study used cytotoxic effects of LL on G. intestinalis including cell cycle progression, determination of reactive oxygen species (ROS), apoptosis/necrosis events and ultrastructural alterations followed by molecular docking approach to determinate its potential as an antiparasitic agent.  Our findings suggest that LL is a neo-clerodane type diterpene with antigiardial potential that induces alterations at the ultrastructural level such as loss of vacuolar presence, and increase in electron-dense granules in G. intestinalis trophozoites by a pronecrotic mechanism involving arrest at S phase and absence of ROS. In addition, molecular docking study suggest that antigiardial effects may be explained in part by the affinity of LL to glycolytic enzyme GdAldRed.  Future systematic works will investigate how LL regulate the expression of glycolytic enzyme GdAldRed and its association with cell cycle, apoptosis/necrosis events and cytoskeleton alterations

Query 7: The reference list not completely fulfill the journal requirement. Please carefully revise it!

Answer: All references were checked and corrected to Pharmaceutical requirement.

Although reviewer comment that “I don't feel qualified to judge about the English language and style” the manuscript was checked by all authors and a native English-speaking colleague.

Reviewer 2 Report

Linearolactone Induces Necrotic-like Death in Giardia duodenalis Trophozoites: Prediction of a Likely Target

In the manuscript, the authors used  Linearolactone to show that LL triggers alterations at the ultrastructural level in G. duodenalis trophozoites. The authors claim that LL induced pronecrotic death and ultrastructural alterations as changes in vacuole abundances, the appearance of perinuclear and periplasmic spaces, and deposition of glycogen granules. In addition, they proposed a pronecrotic mechanism involving arrest at the S phase and the absence of ROS.

Although the authors performed several experiments and obtained some interesting results, such as the absence of ROS, some conclusions were not convincing for this referee. It is not possible to observe the indications mentioned by the authors in the electron microscopy images because the magnification presented is very low. In addition, the authors do not mention how many cell counts were performed to reach these ultrastructural conclusions. On the other hand, the suggestion of cell death by pre-necrosis was not well presented. No image of cells with this behavior was presented.

Comments

Although the species name of Giardia is correct, it was agreed that the most appropriate name would be Giardia intestinalis.

Keywords: linearolactone. Please, remove this keyword since it is already in the title..

Figure 2. The colors do not correspond to the indications.

Figure 4 is a problem. It is not possible to see anything indicated. Therefore, the figures must be presented in higher magnification, and the arrows should be decreased in size.

Line 176-,..” it was foud that,” please correct to “it was found that”

Lines 223-225 “At the trophozoite ultrastructure level, studies carried out with other terpenes (in- comptine A), have reported alterations at the level of the cytoskeleton, decrease in the presence of peripheral vacuoles and increase in glycogen granules in parasites such as Entamoeba histolytica” – reference ?

Lines 231-232: “such as the loss of vacuoles and alterations such as the appearance of irregularly shaped perinuclear and periplasmic spaces, devoid of electron-dense content and areas outlined by membranes with deposition of electron dense granule”- This referee does not agree with this conclusion since the authors did not show clearly these findings.

Line 294: “ Thin sections of 70-500 nm”.Why the authors use 500 nm sections? This is a thick section and it is not indicated for routi

Author Response

Reviewer 2

Query 1: Although the species name of Giardia is correct, it was agreed that the most appropriate name would be Giardia intestinalis.

Answer: In agree with the reviewer in all manuscript Giardia duodenalis was changed by Giardia intestinalis including in the title.

Query 2: Keywords: linearolactone. Please, remove this keyword since it is already in the title.

Answer: linearolactone was deleted in Keywords

Query 3: Figure 2. The colors do not correspond to the indications.

Answer: The Figure 2 was corrected

Query 4: Figure 4 is a problem. It is not possible to see anything indicated. Therefore, the figures must be presented in higher magnification, and the arrows should be decreased in size.

Answer: Figure 4 was changed as Figure 4 and Figure 5 both are presented in high resolution

Query 5: Line 176-,..” it was foud that,” please correct to “it was found that”

Answer: The phrase “it was foud that” in line 176 was corrected as “it was found that”

Query 6: Lines 223-225 “At the trophozoite ultrastructure level, studies carried out with other terpenes (in- comptine A), have reported alterations at the level of the cytoskeleton, decrease in the presence of peripheral vacuoles and increase in glycogen granules in parasites such as Entamoeba histolytica” – reference ?

Answer: Reference was included

Velázquez-Dominguez, J.; Marchat, L.; López-Camarillo, C.; Mendoza-Hernández, G.; Sánchez-Espindola, E.; Calzada, F.; Ortega-Hernández, A. Effect of the sesquiterpene lactone incomptine A in the energy metabolism of E. histolytica. Exp Parasitol. 2013, 135, 503−510.

Query7: Lines 231-232: “such as the loss of vacuoles and alterations such as the appearance of irregularly shaped perinuclear and periplasmic spaces, devoid of electron-dense content and areas outlined by membranes with deposition of electron dense granule”- This referee does not agree with this conclusion since the authors did not show clearly these findings.

Answer: Lines 231-232 was restructured and additional text was included  (see manuscript corrected Lines 287-290, Figure 5)

Query 8:

Line 294: “ Thin sections of 70-500 nm”.Why the authors use 500 nm sections? This is a thick section and it is not indicated for routi

Answer: This part was corrected as “Thin sections of 70 nm”

Although reviewer comment that “English language and style are fine/minor spell check required” the manuscript was checked by all authors and a native English-speaking colleague

Round 2

Reviewer 2 Report

The authors adequately answered the questions and modified what was suggested.